# Online search interest in major depressive disorder: Infodemiology study using the most visited search engine in Japan

Rikako Shimizu[1], Kosuke Ishizuka[1]*, Kagami Kobayakawa[1], Taiju Miyagami[2], Mizue Saita[2], Hirotake Uchikado[3], Tomoki Yamada[4], Tomoya Sakama[1], Akihiko Kusakabe[1], Mitsuyasu Ohta[1], Toshio Naito[2]

1 Department of General Medicine, Yokohama City University School of Medicine, Yokohama, Japan, 2 Department of General Medicine, Juntendo University Faculty of Medicine, Tokyo, Japan, 3 Memory Care Clinic Shonan, Hiratsuka, Japan, 4 Itsuki Clinic Kamariya, Yokohama, Japan

* e103007c@yokohama-cu.ac.jp

## Abstract

Individuals with major depressive disorder (MDD) have an increased risk of suicide and work impairment; therefore, providing appropriate treatment as soon after the onset as possible is critically important. Previous studies on public interest in MDD based on online search trends have addressed temporal changes and regional differences, whereas analyses focusing on the content of search queries remain limited. This study aimed to describe online search trends on MDD in Japan and investigate the relationship between societal perceptions of depression and the symptoms constructs in clinical diagnosis. We conducted a descriptive observational study using the search volume of queries containing "Utsu-byou" (Depression; Major Depressive Disorder in Japanese) from January 2022 to December 2024 using data obtained from Yahoo! JAPAN. We compared the search volume for the top 500 queries on depression each year, and the search volume of queries of symptoms co-searched with depression. These symptoms were classified according to "Component 1—Symptoms and complaints" of the International Classification of Primary Care, 2nd edition (ICPC-2). Furthermore, symptoms classified based on the ICPC-2 were categorized according to whether they corresponded to the diagnostic criteria for MDD in the *Diagnostic and Statistical Manual of Mental Disorders, Fifth Edition* (DSM-5-TR). The search volume for "Depression symptoms" was higher than that for "Depression" alone. Many of the symptoms co-searched with depression corresponded to the diagnostic criteria for MDD in the DSM-5-TR. Specifically, sleep disturbance, which is also frequently observed in clinical practice, was the most frequently co-searched symptom. Public perceptions online of depressive symptoms may demonstrate a certain degree of consistency with clinical diagnostic concepts. Online search data may help clinicians understand how patients recognize and frame depressive symptoms in primary care settings.

**Data availability statement:** The data underlying the results presented in this study are available from Yahoo! JAPAN DS.INSIGHT (Yahoo Japan Corporation). Access to these data is restricted in accordance with the terms and conditions of the data provider and cannot be shared publicly by the authors. Researchers who meet the criteria for access to the data may obtain the data directly from Yahoo! JAPAN DS.INSIGHT (https://datasolution. yahoo-net.jp/service/ds_insight/) by registering for the service and complying with the provider's usage policies. Further information about access to the dataset can be obtained from the Yahoo! JAPAN DS.INSIGHT service website or through Yahoo Japan Corporation. The minimal dataset used in this study consists of aggregated search query volumes for the top 500 queries containing the term "うつ病" obtained through Yahoo! JAPAN DS.INSIGHT for the study period. Researchers can replicate the findings of this study by obtaining the same data from the data provider and following the methods described in the paper. The authors did not have any special access privileges to the data that other researchers would not have.

**Funding:** This work was supported by grants from the Yokohama Foundation for Advancement of Medical Science.

**Competing interests:** The authors have declared that no competing interests exist.

## Introduction

Major depressive disorder (MDD) is one of the most prevalent mental disorders, affecting more than 332 million people worldwide in 2021. Its long-term upward trend highlights its seriousness as a growing global health concern [1]. MDD imposes social and economic burdens on those around affected individuals through suicide and impaired work capacity. Accordingly, timely and appropriate therapeutic intervention after onset is critically important from a societal perspective [2,3].

The presence of undiagnosed and untreated cases of MDD remains a critical issue. A previous study reported in 2017 found that approximately 60% of individuals who met the DSM-IV criteria for MDD in the past 12 months had not received treatment in 12 high-income countries [4]. One of the reasons that individuals with MDD are not diagnosed is their reluctance to seek medical care. Stigma surrounding MDD and mistrust in treatment increase the psychological barriers to consulting a psychiatrist [5]. In particular, Japanese individuals are often described as being reluctant to discuss mental illness with others [6]. In such a sociocultural context, the anonymity of online search may render it a preferable means of seeking information. In addition, from the perspective of healthcare providers, MDD is often underdiagnosed. Although an estimated 7–20% of patients in primary care settings have MDD or depressive symptoms, primary care physicians do not diagnose MDD in 30–50% of patients with this condition [7–9]. One reason why depression is overlooked is that patients tend to present with somatic rather than psychological symptoms. In a multinational study of primary care facilities across 15 countries, 45% to 95% of patients with MDD presented with somatic symptoms [10]. Among patients with MDD and anxiety disorders, physicians recognized 77% of those who presented with psychosocial symptoms, but only 22% of those who presented with somatic symptoms alone. [11].

The level of online interest in MDD among members of the general population remains largely unexplored. More specifically, previous research has largely been confined to examining the association between depression risk and indicators such as temporal fluctuations and geographic variation in search volume using Google Trends [12–14]. In contrast, studies that directly examine the substantive content of search queries remain scarce in part because Google Trends provides limited access to detailed information on the specific content of search queries. Therefore, this study aimed to characterize public interest in MDD in Japan using online search data and to describe the temporal trends in online searches related to MDD in Japan by year during the period 2022–2024. Additionally, we sought to investigate the relationship between societal perceptions of depression and the symptoms constructs in clinical diagnosis.

## Methods

### Data sources

To investigate trends in online searches related to MDD, we obtained the number of search queries containing the term "うつ病" (Utsu-byo, Depression; Major Depressive Disorder) from Yahoo! JAPAN for the period from January 2022 to December

2024. The period from 2022 to 2024 was selected based on considerations of data availability and stability. In this study, "Depression" was used to represent queries searched by the general public, whereas the clinical concept was translated as "Major Depressive Disorder". In Japanese, the term "うつ病" can refer both to depression in a general sense and to MDD as defined in the DSM-5-TR [15]. Therefore, it is not possible to determine which meaning a search query intends, and the term was translated in this way.

Yahoo! JAPAN has maintained the second-largest market share among search engines in Japan for many years, following Google [16]. Yahoo! provides data on the top 500 queries containing specific terms, along with information on the gender and age group of the users who performed these searches. On the other hand, Google Trends, which retrieves data from Google, provides only the regional interest trends for each query, the top 50 related queries, and the top 50 queries with rising search volume. In addition, web audience measurement data indicate that Yahoo! JAPAN continues to be widely used in Japan, with approximately 67.7% of the population using the service at least once per month in 2024, compared with 68.8% for Google during the same period [17,18]. In Japan, individuals under 30 years old use Google almost twice as often as Yahoo! JAPAN, while users aged 40 and above tend to favor Yahoo! JAPAN over Google [19].

Search volume data were retrieved from the server of Yahoo! People on April 12, 2025, using Yahoo! JAPAN DS.INSIGHT, an analytic tool available to registered researchers, that enables analysis of all big behavioral data of Yahoo! Japan, such as keywords, by age, sex, and time period. The user manual is available at https://datasolution.yahoo-net.jp/view/knowledgebase/2205. Data definitions can be accessed from https://datasolution.yahoo-net.jp/view/knowledgebase/post/2596. Yahoo! JAPAN determines users' age and gender based on the information registered in their user accounts. Access to the data used in this research is restricted in accordance with the policies of Yahoo! JAPAN Corporation. The analyses were conducted in compliance with the terms and conditions governing the use of Yahoo! JAPAN DS.INSIGHT.

The internet user volume was defined as the absolute number of internet users in Japan. To define the attributes of the population underlying the search data, we calculated the volume of the internet users in Japan by sex and age group based on the proportion of the internet use from the "Communication Usage Trend Survey" and population data from the "Population Estimates" conducted by the Ministry of Internal Affairs and Communications [20–23]. The "Communications Usage Trend Survey" is a probabilistic survey conducted to assess trends in telecommunications usage, included 39,577 respondents in 2022, 34,196 in 2023, and 37,058 in 2024 [20–22]. The "Population Estimates" provide population estimates by sex and age group based on the national census and other demographic data reflecting population changes [23]. In this study, population data as of October 1 of each year were used.

## Search queries used in the analysis

We assessed the search volume of the top 500 queries containing "Utsu-byou" for each year during the period 2022–2024. Additionally, among the top 500 queries containing "Utsu-byou" during this period, we compared the search volume of queries of symptoms searched with depression. The symptoms were classified according to "Component 1—Symptoms and complaints" of the International Classification of Primary Care, 2nd edition (ICPC-2) [24]. The ICPC-2 focuses on patients' presenting complaints and provides a symptom-based classification system, making it well-suited to the objectives of this study. For the symptom classification, each term was independently evaluated by two researchers (RS and KK), and the results were subsequently compared. Then, an additional evaluation by a third researcher (KI) was performed to minimize observer bias. For example, both "Depression can't sleep" and "Depression just sleeping all day" were classified under "Sleep disturbance", allowing for consistent categorization across slang and variant expressions. As a representative example, Cohen's kappa coefficient was calculated for all 93 symptom queries from 2024, yielding $\kappa = 0.751$, indicating substantial inter-rater agreement and sufficient reproducibility of the classification. A detailed codebook for the classification of search queries into ICPC-2 categories is provided in S1 Table. Furthermore, symptoms categorized based on the ICPC-2 were classified according to whether they were part of the diagnostic criteria for MDD according to the *Diagnostic and Statistical Manual of Mental Disorders, Fifth Edition, Text Revision* (DSM-5-TR).

## Data standardization

The monthly number of searches for each query was calculated by prefecture and adjusted for sex, then standardized into z-scores according to the following formula:

$$z\ score_i = \frac{Search\ volume\ of\ query\ A_i - \overline{Search\ volume\ of\ query\ A}}{Standard\ deviation\ of\ search\ volume\ of\ query\ A}$$

where:

$$\overline{Search\ volume\ of\ query\ A} = \frac{\sum_{i=1}^{n} Search\ volume\ of\ query\ A}{n}$$

and:

$$Standard\ deviation\ of\ search\ volume\ of\ query\ A = \sqrt{\frac{\sum_{i=1}^{n}\left(Search\ volume\ of\ query\ A_i - \overline{Search\ volume\ of\ query\ A}\right)^2}{n-1}}$$

$\overline{Search\ volume\ of\ query\ A}$ represents the mean search volume of query A across all months in the study period.

## Statistical analysis and reporting guidelines

Internet users were stratified by age on the day of the search, sex, and year. Descriptive analyses were performed using Microsoft Excel (Microsoft Corporation, Redmond, WA, USA). The results were reported according to The Strengthening the Reporting of Observational Studies in Epidemiology (STROBE) guidelines.

## Ethics

The requirement for ethical review was waived by the Ethics Committee of Juntendo University Hospital (Tokyo, Japan) because this study was based on a secondary analysis of anonymized pre-existing data. The dataset contained only aggregated search query information and did not include any personally identifiable information. Therefore, informed consent from participants was waived by the ethics committee. The study was conducted in accordance with the principles of the Declaration of Helsinki.

## Results

Table 1 presents the estimated population of the internet users in Japan, derived from the "Communication Usage Trend Survey" and "Population Estimates", alongside the search volume for the query "Depression," broken down by sex and age group [20–23]. Between 2022 and 2024, 50–51% of internet users were male, and the user population was comparatively evenly distributed across the different age groups (Table 1). Of internet users who conducted depression-related searches, 58–60% were women and 40–42% were men. The age group with the highest number of depression-related searches was the 20–29-years age group in 2022 and 2023 (26% and 23%, respectively), and the 50–59-years age group in 2024 (23%) (Table 1).

The most frequently searched term among depression-related searches was "Depression symptoms" with search volumes of 598,000, 509,000, and 504,000, in 2022, 2023, and 2024, respectively (Table 2). The second and third most common queries in each year were "Depression" or "Depression test". The search volume for "Depression" was 122,000, 92,400, and 156,000 in years 2022, 2023, and 2024, respectively, whereas the search volume for "Depression test" during these 3 years was 195,000, 140,000, 109,000, respectively (Table 2).

The most frequently searched queries related to depressive symptoms are shown in Table 3. Symptoms corresponding to the DSM-5-TR diagnostic criteria for MDD are indicated by underlining. "Sleep disturbance" was the most frequently searched symptom in all years. "Feeling depressed", "Feeling/behaving irritable/angry", "Weakness/tiredness

Table 1. Internet search volumes for "Depression" by the sex and age of the user and year (2022–2024).

| | Year 2022 | | | | Year 2023 | | | | Year 2024 | | | |
|---|---|---|---|---|---|---|---|---|---|---|---|---|
| | Internet user[a] | | Search term "Depression"[b] | | Internet user[a] | | Search term "Depression"[b] | | Internet user[a] | | Search term "Depression"[b] | |
| | Volume | (%)[c] | Search volume | (%)[c] | Volume | (%)[c] | Search volume | (%)[c] | Volume | (%)[c] | Search volume | (%)[c] |
| Overall | 99,252,024 | (100%) | 122,000 | (100%) | 99,347,143 | (100%) | 92,400 | (100%) | 105,974,512 | (100%) | 156,000 | (100%) |
| Sex | | | | | | | | | | | | |
| Male | 49,883,262 | (50%) | 49,000 | (40%) | 50,267,774 | (51%) | 37,800 | (41%) | 53,366,438 | (50%) | 66,000 | (42%) |
| Female | 49,368,762 | (50%) | 72,500 | (60%) | 49,099,369 | (49%) | 54,600 | (59%) | 52,571,563 | (50%) | 89,600 | (58%) |
| Age group | | | | | | | | | | | | |
| <20 | 13,467,288 | (14%) | 13,900 | (12%) | 13,376,729 | (13%) | 8,100 | (9%) | 13,121,994 | (12%) | 8,600 | (5%) |
| 20-29 | 12,406,690 | (13%) | 31,900 | (26%) | 12,228,406 | (12%) | 21,400 | (23%) | 12,497,862 | (12%) | 19,800 | (13%) |
| 30-39 | 13,293,144 | (13%) | 22,400 | (19%) | 13,037,003 | (13%) | 17,400 | (19%) | 13,108,784 | (12%) | 22,300 | (14%) |
| 40-49 | 16,718,688 | (17%) | 22,400 | (18%) | 16,143,714 | (16%) | 18,100 | (20%) | 16,066,818 | (15%) | 33,500 | (22%) |
| 50-59 | 16,398,504 | (17%) | 16,500 | (14%) | 16,873,071 | (17%) | 14,600 | (16%) | 17,711,382 | (17%) | 36,500 | (23%) |
| 60-69 | 12,725,846 | (13%) | 7,800 | (6%) | 12,920,129 | (13%) | 6,900 | (7%) | 13,431,462 | (13%) | 20,000 | (13%) |
| ≧70 | 14,231,864 | (14%) | 6,600 | (5%) | 14,788,091 | (15%) | 5,900 | (6%) | 15,493,553 | (15%) | 14,900 | (10%) |

[a]The internet user population was defined as the number of the internet users throughout Japan, calculated from the population of Japan by sex and age group and the corresponding the internet usage rates based on the "Communication Usage Trend Survey" and the "Population Estimates" conducted by the Ministry of Internal Affairs and Communications.

[b]"Search term 'Depression'" indicates the search volume for the query "Depression" and the proportion of searches by each group.

[c]Proportions were calculated using the total number of internet users and the total search volume for the query "Depression" in each year as 100%, respectively.

Table 2. Top 5 online search terms related to major depressive disorder by year (2022-2024).

| Year | 2022 | 2023 | 2024 |
|---|---|---|---|
| No.1 (searches) | Depression symptoms (598,000) | Depression symptoms (509,000) | Depression symptoms (504,000) |
| No.2 (searches) | Depression test (195,000) | Depression test (140,000) | Depression (156,000) |
| No.3 (searches) | Depression (122,000) | Depression (92,400) | Depression test (109,000) |
| No.4 (searches) | Depression how to overcome (82,500) | Adjustment disorder and depression difference (54,500) | Adjustment disorder and depression difference (85,900) |
| No.5 (searches) | Bipolar disorder test (45,500) | Depression how to help (48,600) | Depression how to overcome (51,500) |

The search volume for each query is shown in parentheses. The search queries were originally collected in Japanese and translated into English by the author.

general", "Memory disturbance", "Suicide/suicide attempt" were consistency ranked within the top 10 across all years. All of these symptoms were consistent with the diagnostic criteria of the DSM-5-TR for MDD. In addition, among the symptoms not included in the DSM-5-TR diagnostic criteria, "Headache" ranked first in 2023 and 2024, excluding the residual category.

**Table 3. Top 20 search terms and symptoms in online searches related to major depressive disorder by year (2022–2024).**

| Year | 2022 | 2023 | 2024 |
|---|---|---|---|
| Number of searches for Depression (searches; %) | Number of searches for "Depression" (122,000; 100%) | Number of searches for "Depression" (92,400; 100%) | Number of searches for "Depression" (156,000; 100%) |
| No.1 (searches; %) | Sleep disturbance (36,710; 30.1%) | Sleep disturbance (24,050; 26.0%) | Sleep disturbance (21,380; 13.7%) |
| No.2 (searches; %) | Feeling depressed (10,150; 8.3%) | Feeling/behaving irritable/angry (13,570; 14.7%) | Feeling depressed (12,280; 7.9%) |
| No.3 (searches; %) | Feeling/behaving irritable/angry (10,120; 8.3%) | Feeling depressed (11,960; 12.9%) | Limited function/disability NOS (11,860; 7.6%) |
| No.4 (searches; %) | Memory disturbance (9,290; 7.6%) | Limited function/disability NOS (11,360; 12.3%) | Feeling/behaving irritable/angry (10,600; 6.8%) |
| No.5 (searches; %) | Suicide/suicide attempt (8,680; 7.1%) | Memory disturbance (7,480; 8.1%) | Weakness/tiredness general (7,650; 4.9%) |
| No.6 (searches; %) | Limited function/disability NOS (7,650; 6.3%) | Weakness/tiredness general (7,390; 8.0%) | Psychological symptom/complaint other (5,900; 3.8%) |
| No.7 (searches; %) | Weakness/tiredness general (6,540; 5.4%) | Suicide/suicide attempt (5,500; 6.0%) | Memory disturbance (5,710; 3.7%) |
| No.8 (searches; %) | Feeling anxious/nervous/tense (5,280; 4.3%) | Psychological symptom/complaint other (5,160; 5.6%) | Suicide/suicide attempt (4,970; 3.2%) |
| No.9 (searches; %) | Psychological symptom/complaint other (4,980; 4.1%) | Loss of appetite (4,300; 4.7%) | Headache (4,400; 2.8%) |
| No.10 (searches; %) | Headache (4,800; 3.9%) | Headache (4,190; 4.5%) | Loss of appetite (4,200; 2.7%) |
| No.11 (searches; %) | Loss of appetite (4,200; 3.4%) | Feeling anxious/nervous/tense (3,100; 3.4%) | Fever (3,010; 1.9%) |
| No.12 (searches; %) | Nausea (2,900; 2.4%) | Palpitations/awareness of heart (2,520; 2.7%) | Vertigo/dizziness (2,640; 1.7%) |
| No.13 (searches; %) | Palpitations/awareness of heart (2,780; 2.3%) | Vertigo/dizziness (2,520; 2.7%) | Excessive appetite (2,340; 1.5%) |
| No.14 (searches; %) | Fever (2,630; 2.2%) | Nausea (2,500; 2.7%) | Nausea (2,300; 1.5%) |
| No.15 (searches; %) | Vertigo/dizziness (2,600; 2.1%) | Weight loss (2,200; 2.4%) | Weight loss (2,090; 1.3%) |
| No.16 (searches; %) | Weight loss (1,940; 1.6%) | Fever (2,030; 2.2%) | Weight gain (1,800; 1.2%) |
| No.17 (searches; %) | Weight gain (1,900; 1.6%) | Weight gain (1,700; 1.8%) | Palpitations/awareness of heart (1,800; 1.2%) |
| No.18 (searches; %) | Tinnitus, ringing/buzzing ear (1,690; 1.4%) | Excessive appetite (1,600; 1.7%) | Low back symptom/complaint (1,460; 0.9%) |
| No.19 (searches; %) | Excessive appetite (1,600; 1.3%) | Low back symptom/complaint (1,530; 1.7%) | Limited function/disability (1,440; 0.9%) |
| No.20 (searches; %) | Shortness of breath/dyspnoea (1,200; 1.0%) | Shortness of breath/dyspnoea (1,000; 1.1%) | Shortness of breath/dyspnoea (960; 0.6%) |

The underlined terms indicate symptoms of major depressive disorder listed in the *Diagnostic and Statistical Manual of Mental Disorders, Fifth Edition, Text Revision* (DSM-5-TR). The search volume for each query is shown in parentheses. The search queries were originally collected in Japanese and translated into English by the author.

## Discussion

From 2022 to 2024, "Depression symptoms" consistently had the highest search volume, surpassing that of "Depression" alone. This suggests that members of the general public were more interested in the symptoms of MDD, rather than the disorder itself. The third most searched query was "Depression test." which probably indicates an interest in self-diagnosis or self-diagnosis tools. This strong interest in depressive symptoms and self-diagnosis, suggest that a high proportion of the individuals who performed the searches may have been looking for an explanation of their own symptoms when conducting the online searches.

In recent years, websites have become available that enable self-diagnosis of MDD. A web-based screening tool has been reported to demonstrate relatively high accuracy, with a sensitivity of 0.79 and a specificity of 0.89, suggesting that online self-diagnosis can be effective as a screening approach [25]. Furthermore, such search behaviors could contribute to improving doctor-patient relationships and encourage individuals with symptoms of MDD to consider seeing a health professional [26,27]. These websites can therefore be regarded as useful tools for supporting medical care. Conversely, self-diagnosis may foster misconceptions or bolster preconceived notions, which can create barriers to accepting medical advice and may also reinforce patients' assuming a sick role [28,29].

Based on the finding that searches related to MDD symptoms was the most frequent type of MDD-related online search, we examined which specific symptoms were the most frequently searched. The symptoms explicitly listed in the DSM-5-TR as diagnostic criteria for MDD, classified according to ICPC-2, are feeling depressed, weight gain, weight loss, excessive appetite, loss of appetite, sleep disturbance, feeling/behaving irritable/angry, general weakness/tiredness, memory disturbance, and suicide/suicide attempt [17]. As shown in Table 3, many of the high-volume search queries related to depressive symptoms corresponded to the symptoms listed in the DSM-5-TR criteria. Thus, many of the symptoms of MDD that attracted high online interest correspond to those included in the DSM-5-TR diagnostic criteria. This finding suggests that public awareness of depressive symptoms may be partially aligned with clinical diagnostic concepts.

During the study period, the symptom query with the highest search volume was "Sleep disturbance." Notably, this ranked above "Feeling depressed," which is a core symptom in the DSM-5-TR diagnostic criteria for MDD. Previous studies have reported that sleep disturbance is present in more than 75% of patients with MDD, and it is a common symptom that prompts healthcare-seeking behavior in primary care settings [30–32]. The finding that sleep disturbance attracted substantial online interest as a symptom of MDD is consistent with the frequent occurrence of sleep disturbance in individuals with MDD.

Feeling/behaving irritable/angry, weakness/tiredness general, and memory disturbance are frequently searched and have been reported to be present in over 40% of individuals with MDD [33–36]. These symptoms are consistent with the DSM diagnostic criteria, and, similar to sleep disturbance, the findings suggest that symptoms frequently observed in clinical practice also attract considerable attention online.

Headache was the query with the highest search volume in 2023 and 2024 among somatic symptoms not listed in the DSM-5-TR diagnostic criteria, excluding those classified as residual symptoms. Approximately 45% of individuals with MDD experience headaches and a significant association between MDD and headaches has been reported [37–39]. Therefore, headache appears to be one of the more commonly experienced somatic symptoms among individuals with MDD. Although headache is not listed as one of the DSM-5-TR diagnostic criteria, more attention should be given to this symptom in clinical practice.

The findings of this study suggest that the general public may search for symptoms related to themselves or those around them in connection with depression, and that the symptoms of greatest interest are consistent with the diagnostic criteria outlined in the DSM-5-TR. This alignment illuminates social perceptions of depression as reflected in online search behavior. Being mindful of symptoms that attracted high levels of attention online may help primary care physicians understand patient-recognized symptom framings of depression, particularly when patients present with non-specific or somatic complaints. Future research should investigate the relationship between the presence of depression or help-seeking behaviors and search activity to further strengthen these interpretations.

This study has some limitations. First, it used data extracted from Yahoo! JAPAN as the sole source of search data and did not include data from other search engines. Therefore, the data used in this study may not reflect the search behaviors of all internet users in Japan, and differences between search engines in user attributes or sampling methods may have influenced the results. Second, owing to the study design, the results do not reflect the level of interest in MDD among population subgroups that do not use the internet such as older adults. Hence, the findings may not be generalizable to the overall population of Japan. Third, biases may arise during data extraction and classification. To minimize observer bias, evaluations were independently performed by two assessors (RS, KK), with an additional assessment by a third assessor (KI). Fourth, aggregation of queries during the classification may have influenced the interpretation of the results. Fifth, the search data used in this study likely includes searches performed not only by individuals experiencing symptoms of MDD but also by people in various roles, such as family members or other close contacts, healthcare professionals, and researchers. The searchers may also include individuals with other conditions who mistakenly believe that they have MDD. In addition, increased media coverage of celebrities or health-related information about depression may lead to higher search volumes driven purely by public interest. Therefore, these search trends should not be assumed to reflect the characteristics of individuals with MDD. Sixth, selection bias may exist based on the severity of MDD. In severe MDD, symptoms such as psychomotor retardation may make performing online searches difficult. Therefore, individuals with severe MDD may be underrepresented. Seventh, the data on age and sex were obtained from the login information of Yahoo! Users. However, this information may not necessarily correspond to the age and sex of the users doing the searches. Internet users may have provided inaccurate information on their age or sex when creating an account, and someone other than the account holder may have used the account to conduct searches. Eighth, there is a possibility of bias due to orthographic variants. In Japanese, "Utsu" can be written in both hiragana and kanji, resulting in two forms, "うつ" and "鬱", and in this study we adopted the form "うつ". The kanji character "鬱" was added to the list of commonly used kanji in 2010 [40]. In practice, however, the form "うつ病" is more widely used in everyday contexts and is more familiar to the general public. The Japanese version of the DSM-5 and DSM-5-TR also adopts "うつ病", indicating that it is the standard term in medical settings [17]. For these reasons, we selected "うつ病" to ensure terminological and clinical consistency. Variations in orthography may be associated with differences in demographic characteristics, such as gender or age group. However, because the data obtained in this study included only queries containing "うつ病", potential differences related to alternative forms were not examined and should be addressed in future research.

## Conclusion

This study describes trends in online interest in MDD in Japan using search behavior data. Interest in the symptoms of depression was highest, suggesting that the general public may be searching for information under the assumption that their own symptoms are associated with MDD. In addition, because many of the symptoms which attracted the highest level of interest corresponded to the DSM-5-TR diagnostic criteria for MDD, the searchers may have included patients diagnosed with MDD. Specifically, sleep disturbance, which are frequently observed in clinical settings, was the most frequently searched symptom. These findings illuminate social perceptions of depression and its symptoms as reflected in online search behavior. Public perceptions online of depressive symptoms may demonstrate a certain degree of consistency with clinical diagnostic concepts. Online search data may help clinicians understand how patients recognize and frame depressive symptoms, particularly those that overlap with clinical diagnostic concepts. Further research is needed to determine the extent to which such behavior reflects clinical condition.

## Supporting information

**S1 Table. Codebook for classification of symptoms-related queries based on ICPC-2.**
(PDF)

## Author contributions

**Conceptualization:** Rikako Shimizu.

**Data curation:** Rikako Shimizu, Kosuke Ishizuka, Kagami Kobayakawa, Taiju Miyagami.

**Formal analysis:** Rikako Shimizu, Kosuke Ishizuka, Kagami Kobayakawa, Taiju Miyagami.

**Funding acquisition:** Kosuke Ishizuka.

**Investigation:** Rikako Shimizu, Kosuke Ishizuka, Kagami Kobayakawa, Taiju Miyagami.

**Methodology:** Rikako Shimizu, Kosuke Ishizuka, Kagami Kobayakawa, Taiju Miyagami.

**Project administration:** Toshio Naito.

**Resources:** Rikako Shimizu, Kosuke Ishizuka, Kagami Kobayakawa.

**Software:** Rikako Shimizu.

**Supervision:** Mitsuyasu Ohta.

**Validation:** Kosuke Ishizuka.

**Visualization:** Rikako Shimizu, Kagami Kobayakawa.

**Writing – original draft:** Rikako Shimizu.

**Writing – review & editing:** Kosuke Ishizuka, Kagami Kobayakawa, Taiju Miyagami, Mizue Saita, Hirotake Uchikado, Tomoki Yamada, Tomoya Sakama, Akihiko Kusakabe, Mitsuyasu Ohta, Toshio Naito.

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
