## [Decision Letter · Decision Letter 0]

3 Feb 2026

PONE-D-25-64954Online Search Interest in Major Depressive Disorder: Infodemiology Study Using the Most Visited Search Engine in JapanPLOS One Dear Dr. Ishizuka,

Thank you for submitting your manuscript to PLOS ONE. After careful consideration, we feel that it has merit but does not fully meet PLOS ONE’s publication criteria as it currently stands. Therefore, we invite you to submit a revised version of the manuscript that addresses the points raised during the review process.

Thank you for submitting your manuscript to PLOS ONE. After careful consideration of two expert reviews, I am recommending major revision before your manuscript can be considered for publication.

Both reviewers recognize the potential value of your taxonomic approach to classifying MDD-related search queries. However, they have raised critical concerns that must be addressed, particularly regarding:

1. Over-interpretation of findings (Critical)

Both reviewers and I are concerned about the risk of conflating public search interest with clinical prevalence or diagnostic capability. As Reviewer 1 notes: "Search volume reflects information-seeking behavior rather than clinical status."

Required changes:

Clearly distinguish throughout the manuscript between information-seeking behavior and actual MDD prevalence

Remove or substantially qualify claims about "proactive detection" unless validated against actual diagnosis data

Reframe implications for clinical practice more cautiously

2. Unclear research objective (Critical)

Reviewer 2 identifies fundamental ambiguity about whether this study aims to:

Predict MDD diagnosis rates

Improve diagnostic understanding

Characterize public interest

Reduce stigma

Required changes:

State a clear, focused research question in the abstract and introduction

Align the discussion strictly with this objective

Remove tangential discussion points not supported by your analysis

3. Methodological transparency (Critical)

Both reviewers request essential methodological details:

Required changes:

Report inter-rater reliability for symptom classification (Cohen's kappa, percent agreement)

Justify selection of "Utsu-byou" as the search term

Test and report results for common variants (うつ病 vs 鬱病)

Verify Yahoo! JAPAN market share claim with appropriate citation

Provide rationale for Japan and 2022-2024 timeframe

4. Literature review (Major)

Reviewer 1 notes the background "does not sufficiently review prior work on using online search behavior to characterize public interest in mental health conditions."

Required changes:

Strengthen literature review on mental health infodemiology

Situate your contribution within existing research

Acknowledge limitations identified in prior Google Trends studies (see Reviewer 2's citations)

5. Additional concerns

Please address all specific comments from both reviewers, including:

Data quality and representativeness (Yahoo! vs. Google users)

Table clarity and reporting proportions

Spelling errors and presentation issues

References to publicly accessible documentation

Important Note on Scope:

Given the concerns about over-interpretation, I strongly encourage you to frame this as a descriptive infodemiology study of public information-seeking behavior rather than a clinical detection tool. If you wish to make claims about early detection or clinical utility, you must provide validation against actual MDD diagnosis data, which your current study does not include.

We look forward to receiving your revised manuscript.

Kind regards,

Rei Akaishi

Academic Editor

PLOS One

Journal Requirements:

In your Methods section, please include additional information about your dataset and ensure that you have included a statement specifying whether the collection and analysis method complied with the terms and conditions for the source of the data.

“This work was supported by grants from the Yokohama Foundation for Advancement of Medical Science.”

“This work was supported by grants from the Yokohama Foundation for Advancement of Medical Science.”

Additional Editor Comments:

Dear Authors,

Thank you for submitting your manuscript to PLOS ONE. After careful consideration of two expert reviews, I am recommending major revision before your manuscript can be considered for publication.

Both reviewers recognize the potential value of your taxonomic approach to classifying MDD-related search queries. However, they have raised critical concerns that must be addressed, particularly regarding:

1. Over-interpretation of findings (Critical)

Both reviewers and I are concerned about the risk of conflating public search interest with clinical prevalence or diagnostic capability. As Reviewer 1 notes: "Search volume reflects information-seeking behavior rather than clinical status."

Required changes:

Clearly distinguish throughout the manuscript between information-seeking behavior and actual MDD prevalence

Remove or substantially qualify claims about "proactive detection" unless validated against actual diagnosis data

Reframe implications for clinical practice more cautiously

2. Unclear research objective (Critical)

Reviewer 2 identifies fundamental ambiguity about whether this study aims to:

Predict MDD diagnosis rates

Improve diagnostic understanding

Characterize public interest

Reduce stigma

Required changes:

State a clear, focused research question in the abstract and introduction

Align the discussion strictly with this objective

Remove tangential discussion points not supported by your analysis

3. Methodological transparency (Critical)

Both reviewers request essential methodological details:

Required changes:

Report inter-rater reliability for symptom classification (Cohen's kappa, percent agreement)

Justify selection of "Utsu-byou" as the search term

Test and report results for common variants (うつ病 vs 鬱病)

Verify Yahoo! JAPAN market share claim with appropriate citation

Provide rationale for Japan and 2022-2024 timeframe

4. Literature review (Major)

Reviewer 1 notes the background "does not sufficiently review prior work on using online search behavior to characterize public interest in mental health conditions."

Required changes:

Strengthen literature review on mental health infodemiology

Situate your contribution within existing research

Acknowledge limitations identified in prior Google Trends studies (see Reviewer 2's citations)

5. Additional concerns

Please address all specific comments from both reviewers, including:

Data quality and representativeness (Yahoo! vs. Google users)

Table clarity and reporting proportions

Spelling errors and presentation issues

References to publicly accessible documentation

Important Note on Scope:

Given the concerns about over-interpretation, I strongly encourage you to frame this as a descriptive infodemiology study of public information-seeking behavior rather than a clinical detection tool. If you wish to make claims about early detection or clinical utility, you must provide validation against actual MDD diagnosis data, which your current study does not include.

Reviewers' comments:

Reviewer's Responses to Questions

**Comments to the Author**

1. Is the manuscript technically sound, and do the data support the conclusions?

Reviewer #1: Partly

Reviewer #2: Partly

2. Has the statistical analysis been performed appropriately and rigorously? 

Reviewer #1: Yes

Reviewer #2: I Don't Know

3. Have the authors made all data underlying the findings in their manuscript fully available?

Reviewer #1: Yes

Reviewer #2: Yes

4. Is the manuscript presented in an intelligible fashion and written in standard English?

Reviewer #1: Yes

Reviewer #2: Yes

5. Review Comments to the Author

Reviewer #1: General comment:

This descriptive study uses Yahoo! JAPAN search queries (2022–2024) to characterize public interest in major depressive disorder (“Utsu-byou”), focusing on symptoms co-searched with MDD and classifying them via ICPC-2 with mapping to DSM-5-TR criteria. The taxonomy-based classification is a strength and could be useful for infodemiology and health communication.

However, I have some concerns that should be addressed before publication.

Major Comments:

1. (L66) "The level of interest in MDD among members of the general population remains largely unexplored and research on public interest in MDD through online search behavior is limited."

The current background does not sufficiently review prior work on using online search behavior to characterize public interest in mental health conditions. Please strengthen the literature review and clarify the specific gap this study addresses.

2. (L68) "Therefore, this study aimed to characterize public interest in MDD in Japan using online search data and to describe the temporal trends in online searches related to MDD in Japan by year during the period 2022 to 2024."

Please provide a clearer rationale for focusing on Japan and the 2022–2024 period. As written, the choice of country and time window feels arbitrary. Briefly explain whether this is driven by data availability, contextual factors, policy/health system considerations, or specific events that make this period particularly relevant.

3. (L75) "the term “Utsu-byou” (the Japanese term for MDD)"

Japanese queries have common orthographic variants (e.g., うつ病 vs 鬱病) and may also be used colloquially without implying a clinical diagnosis. Please clarify whether variants were tested and/or included, and consider sensitivity analyses comparing results across key variants. In addition, because “Utsu-byou” searches may include substantial noise, please discuss how this could affect the findings and, if feasible, provide the exact Japanese search terms used (e.g., in a supplement).

4. (L76) "Yahoo! JAPAN is the most frequently used search engine in Japan,"

This statement should be carefully verified and appropriately framed. Metrics such as site visits to yahoo.co.jp (a large portal including news, weather, mail, etc.) are not equivalent to search engine market share. If the intent is to justify platform representativeness, please use a metric directly relevant to search usage (e.g., search engine market share) and revise the wording accordingly. Please cite an appropriate source.

Ideally, provide a supplementary table listing the symptom terms (and representative raw queries) with final ICPC-2 labels and DSM alignment.

5. (L88) "Search queries used in the analysis"

Without a codebook (decision rules) and examples, readers cannot evaluate how raw queries were mapped to symptom terms and then to ICPC-2 categories—especially for slang, spelling variants, multi-term queries, and ambiguous expressions.

Moreover, because two coders independently classified terms with third-party adjudication, please report inter-rater reliability (e.g., Cohen’s kappa and/or percent agreement) and the proportion of disagreements requiring adjudication.

6. (L173) "Table 3. Top 20 search terms and symptoms in online searches related to major depressive disorder by year (2022–2024)."

Please also report proportions to enable valid comparisons across years.

7. (L221) "Given that the diagnostic criteria of the DSM-5-TR are based on a statistical classification of clinically observed symptoms and signs, individuals searching online for

symptoms of MDD may include patients diagnosed with MDD."

⇒Search volume reflects information-seeking behavior rather than clinical status; therefore, findings should be interpreted as public interest in symptom-related concepts linked to “Utsu-byou,” not as evidence of symptom prevalence or diagnosis. Please revise the wording accordingly to avoid potential misunderstanding.

Minor Comments:

1. (L83) "https://datasolution.yahoo-net.jp/view/knowledgebase/641."

If this resource requires a login or is not publicly accessible, it may limit reproducibility for readers. Please provide a publicly accessible reference, or alternatively describe the relevant details in the manuscript or a supplementary appendix.

Reviewer #2: The manuscript “Online Search Interest in Major Depressive Disorder: Infodemiology Study Using the Most Visited Search Engine in Japan” sets out to characterize public interest in Major Depressive Disorder (MDD) in Japan using Yahoo! search trends from 2022 to 2024. The authors compare the most frequent terms co-searched with queries for major depressive disorder (MDD) to official classifications of MDD symptoms. Internet search data offers promising opportunities to measure public interest and public health, and we still need a clearer understanding of when and how researchers can meaningfully apply these data. I applaud the authors for advancing our understanding in this direction. However, I have two major concerns with the paper at hand regarding (1) the study goal and relevance, (2) the study’s empirical approach, and (3) the presentation of results. I elaborate on these points, along with several more minor comments, below. I hope that my comments are constructive and helpful to the authors in further strengthening their research.

(1) Study goal and relevance: The authors should state the study’s research question, objectives, and scientific and societal relevance more clearly from the outset. In the introduction, the authors note that major depressive disorder (MDD) is frequently underdiagnosed or diagnosed with delay and state that the study aims “to characterize public interest in MDD” (p. 3, line 68). However, it remains unclear how the goal of characterizing public interest in MDD relates to the societal problem of delayed MDD diagnoses. It is also not specified in which contexts and for whom such a characterization would be useful and what the search interest in MDD over time would tell us.

The conclusion further states that “focusing on these symptoms may enable primary care physicians to proactively identify cases of MDD” (p. 15, lines 308 to 209). Given that the classification of Yahoo! searches is based on the official classification of MDD symptoms, it remains unclear, however, whether this approach offers information beyond what medical practitioners already rely on clinical practice.

Throughout the manuscript, it is confusing whether the study’s goal is to assess whether online search data could function as an early detection system for MDD by predicting official diagnosis rates, or whether it seeks to improve diagnostic understanding by identifying the most frequently searched MDD symptoms. Clarifying the overarching objective of the study from the outset would substantially strengthen the manuscript.

The discussion should then be more tightly aligned with this clarified objective. Currently, the discussion contains statements that do not reflect any of the goals hinted at in the introduction. For instance, the authors state that “promoting acceptance of MDD among patients and those around them is essential” (p. 11, lines 206 to 208), suggesting an additional goal related to stigma reduction. However, the manuscript does not demonstrate how the presented findings inform or advance this objective. Similarly, the statement that “when disseminating information about diseases, ensuring the accuracy of the content is important” (p.12, lines 236 to 238) is connected to the empirical analysis, as the study does not examine the content of information presented to users following their Yahoo! searches. Overall, the authors must specify the broader goal of their research, focus their abstract, introduction, and discussion on this goal, and streamline their statements in these chapters accordingly.

Finally, the implications of the study for existing and future research are not sufficiently stated, leaving it unclear how researchers might build on the reported findings. The abstract currently notes that “research on public interest in MDD based on online search trends is limited” (p. 2, lines 33 to 34) but does not discuss whether prior studies have attempted to predict MDD using digital trace data or how successful such efforts have been. The abstract and introduction should more clearly situate the study and its contribution within the existing literature, while the discussion should explicitly outline the study’s implications for future research.

(2) Empirical approach: For other Internet search data sources such as Google Trends, previous research has shown that the selection of appropriate search terms poses substantial challenges for establishing internal (measurement) validity (e.g., Mellon 2013, 2014; Hölzl et al. 2025). Similar concerns are likely to apply to the use of Yahoo! search data. Individuals interested in or concerned about MDD may not necessarily search for the term “Utsu-byou,” but instead use other terms. The authors thus need to explain their search term selection of “Utsu-byou” and make a strong case for why people would search for exactly this term when concerned about MDD. In this context, it would be helpful to clarify whether “Utsu-byou” is a commonly used term in Japan.

The classification of search terms co-occurring with MDD-related queries into MDD symptoms represents a very valuable component of the study’s empirical approach. However, the description of the coding procedure would benefit from further clarification. Specifically, the authors note that “an additional evaluation by a third researcher (KI) was performed to minimize observer bias” (p. 5, lines 96–97), but it is not specified how this evaluation was conducted. Typically, reducing observer bias would involve independent coding of a subset of the data by multiple researchers, the calculation of interrater reliability, and the joint resolution of any discrepancies in the coding. The authors should clarify whether multiple coders independently classified a subset of search terms and, if so, report the corresponding interrater reliability. If the third researcher’s role differed from this standard procedure, further explanation is needed.

The manuscript would also benefit from a clearer justification of the case selection. The authors should explain why Japan constitutes a particularly relevant or informative context for their research purposes. Similarly, they should discuss why they selected the two-year analysis period from 2022 to 2024. The authors need to make a case for why the search trends for these two years is particularly interesting for their research purposes.

Finally, the authors should address the quality of the Yahoo! search data. Providing information on data coverage, reliability, and any known limitations of Yahoo!’s search data would strengthen confidence in the empirical findings.

(3) Presentation of results: In the description of the results, the authors currently simply list the frequencies of search queries (e.g., p. 7, line 140, to p. 9, line 175). The authors should focus more on telling a story that aligns with the research’s goal on how these searches relate to the detection of MDDs or the improvement of diagnostics. If the research goal lies in prediction of MDD instead of improving diagnostics, the authors need to add predictions of official statistics of MDD from the Yahoo! searches. More generally, the description and discussion of findings feel quite lengthy and should be condensed.

Table 1 and 2 would also very much benefit from a clarification on the absolute numbers of search volume. For instance, it is not explained what the numbers of the search volume in the column “Internet user” mean and how they have been calculated. The table note should explain the metric. In general, the table notes should be more informative so that readers are able to understand the text without the table and vice versa.

Some minor remarks:

Abstract: In the abstract, the results are described in too much detail, relatively little attention is given to the broader implications of the findings. It would be helpful for readers to grasp from the abstract what the study’s implications are.

p. 3, 51: The term “Major Depressive Disorder” should be defined somewhere. Is MDD the clinical term for what lay people understand under the term depression?

p. 3, 57-59: When discussing this study, it should be stated when the study was conducted and for which regions the results apply.

p. 4, 76ff.: The authors cite a source showing the number of Yahoo! users compared to Google users in Japan in 2021. Has the proportion of Yahoo! users to Google users changed within the last five years? As Google users have also been a substantially large group in 2021, the authors should discuss in the discussion chapter how Yahoo! and Google users differ from each other, i.e., whether there are any systematic differences between the two groups.

p. 4, 82: It should already be added here that Yahoo! gets the information on age and gender from the users’ profiles.

p. 4, 83: As only registered users have access to the user manual, it would be very helpful if the authors would provide some openly accessible documentation of Yahoo! JAPAN DS.INSIGHT so that readers can better inform themselves about the tool.

p. 4, 84ff.: The authors should explain for which research purposes they additionally collected survey data. They also need provide a citation for the survey and provide more information on the survey, such as the field period and number of respondents. Similarly, they need to outline what the “Population Estimates” by the Ministry of Internal Affairs and Communications are, why they are needed for their research purposes, and correctly cite this data source, as well.

p. 4, 90: The authors state that they compared the search volume of queries that also include symptoms in the query. To get a better overall picture of the content of MDD queries, it would be helpful if the authors could also provide what the other search terms searched along with MDD were that are not symptoms.

p. 5, 105: In the formula for the calculation of the z-scores, does the average search volume of query A relate to the average across all months in the data collection period? If yes, this should be clearly denoted in the formula.

p. 6, 125f.: In “50–51% of internet users were male, and the user population was comparatively evenly distributed across the different age groups”, it is unclear whether the numbers reported relate to the survey data, the “Population Estimates” by the Ministry of Internal Affairs and Communications, or the Yahoo! users only.

p. 7, 135f.: The definition of the internet user population from the table note should be included in the description of the data sources as well (p. 4, lines 84-86).

p. 8, 168: There potentially is a spelling error in “psychological symptom/complt other”.

p. 11, 209-2020: This paragraph could be very much condensed by simply stating that “many of the symptoms of MDD that attracted high online interest correspond to those included in the DSM-

5-TR diagnostic criteria” and just providing one or two examples for the corresponding symptoms.

p. 12, 247: There seems to be a spelling mistake in “Irritability or uncreased reactivity”.

p. 14, 281: The authors should very briefly discuss here who the population subgroups are that do not use the Internet in Japan, for instance, older individuals.

p. 14, 286 – 291: The authors nicely discuss several reasons why also individuals that do not suffer from MDD might nevertheless search for MDD. To strengthen their discussion of this limitation, they should add that increased public awareness and media attention to MDD and specific MDD symptoms might also lead to more people searching for the topic, simply out of interest without any personal connection to the issue. A good example for this issue is also Google Flu Trends (e.g., Lazer et al. 2014) which was a service provided by Google until 2015 to predict influenza-like illnesses by using the Google search volume for flu symptoms. However, Google Flu Trends largely overestimated influenza outbreaks compared to official data, in parts, because individuals that did not have the flu also googled flu symptoms.

References cited in the review:

Hölzl, Johanna, Florian Keusch, and Christoph Sajons. 2025. “The (Mis)Use of Google Trends Data in the Social Sciences - A Systematic Review, Critique, and Recommendations.” Social Science Research 126:103099. doi:10.1016/j.ssresearch.2024.103099.

Lazer, David, Ryan Kennedy, Gary King, and Alessandro Vespignani. 2014. “The Parable of Google Flu: Traps in Big Data Analysis.” Science 343(March 14):1203–5.

Mellon, Jonathan. 2013. “Where and When Can We Use Google Trends to Measure Issue Salience?” PS: Political Science & Politics 46(2). doi:10.1017/S1049096513000279.

Mellon, Jonathan. 2014. “Internet Search Data and Issue Salience: The Properties of Google Trends as a Measure of Issue Salience.” Journal of Elections, Public Opinion and Parties 24(1):45–72. doi:10.1080/17457289.2013.846346.

6. PLOS authors have the option to publish the peer review history of their article (what does this mean?). If published, this will include your full peer review and any attached files.

Reviewer #1: No

Reviewer #2: No

---

## [Author Response · Author response to Decision Letter 1]

18 Mar 2026

March 12, 2026

Dear Dr. Akaishi,

Title: Online Search Interest in Major Depressive Disorder: Infodemiology Study Using the Most Visited Search Engine in Japan

Reference number: PONE-D-25-64954

Dear Dr. Akaishi,

Thank you for your e-mail of February 3, 2026, regarding our manuscript, “Online Search Interest in Major Depressive Disorder: Infodemiology Study Using the Most Visited Search Engine in Japan”, and for the valuable comments of the reviewers. I have attached our revised manuscript, as well as a point-by-point response to the reviewers’ comments.

We hope that the revised manuscript contains suitable responses to the comments, and we think that it has been significantly improved over the previous submission. We trust that our manuscript is now suitable for publication in PLOS ONE.

Thank you in advance for your kind consideration of our work.

Sincerely yours,

Kosuke Ishizuka, MD, PhD

Department of General Medicine, Yokohama City University School of Medicine

3-9 Fukuura, Kanazawa-ku, Yokohama-city, Kanagawa pref. Japan

Tel. +81-45-787-2800

Fax. +81-45-350-2728

E-Mail: e103007c@yokohama-cu.ac.jp

RESPONSES TO REVIEWER 1:

We sincerely thank the reviewer for insightful comments that have helped us to improve our paper. We believe that the revised version adequately addresses all concerns raised, and we think that it has been significantly improved over the previous submission. We trust that our manuscript is now suitable for publication in PLOS ONE.

NOTE

We highlighted changes of significant issues in yellow (please see “Revised Manuscript with Track Changes”).

Comments:

This descriptive study uses Yahoo! JAPAN search queries (2022–2024) to characterize public interest in major depressive disorder (“Utsu-byou”), focusing on symptoms co-searched with MDD and classifying them via ICPC-2 with mapping to DSM-5-TR criteria. The taxonomy-based classification is a strength and could be useful for infodemiology and health communication.

However, I have some concerns that should be addressed before publication.

Response:

Thank you for your general comments. In accordance with your suggestions, we have revised the manuscript. We have addressed each of the points you raised.

Major comments:

1. (L66) "The level of interest in MDD among members of the general population remains largely unexplored and research on public interest in MDD through online search behavior is limited."

The current background does not sufficiently review prior work on using online search behavior to characterize public interest in mental health conditions. Please strengthen the literature review and clarify the specific gap this study addresses.

Response:

We agree with your assessment. Along with your suggestion, we have revised the Introduction to include a more detailed review of prior research and to clarify the novelty of the present study. We also modified the Abstract accordingly.

Changes:

Previous studies on public interest in MDD based on online search trends have addressed temporal changes and regional differences, whereas analyses focusing on the content of search queries remain limited. (Abstract; lines 32 to 34.)

More specifically, previous research has largely been confined to examining the association between depression risk and indicators such as temporal fluctuations and geographic variation in search volume using Google Trends [10-12]. In contrast, studies that directly examine the substantive content of search queries remain scarce in part because Google Trends provides limited access to detailed information on the specific content of search queries. (Introduction; lines 73 to 78.)

2. (L68) "Therefore, this study aimed to characterize public interest in MDD in Japan using online search data and to describe the temporal trends in online searches related to MDD in Japan by year during the period 2022 to 2024."

Please provide a clearer rationale for focusing on Japan and the 2022–2024 period. As written, the choice of country and time window feels arbitrary. Briefly explain whether this is driven by data availability, contextual factors, policy/health system considerations, or specific events that make this period particularly relevant.

Response:

We thank the reviewer for this valuable comment. We agree that the rationale for selecting the country and study period was insufficiently explained. We have added an explanation in the Introduction regarding the social background supporting our focus on Japan. In addition, we have clarified in the Methods section the reasons for selecting the three-year period, including data availability and the temporal stability of the data.

Changes:

In particular, Japanese individuals are often described as being reluctant to discuss mental illness with others [6]. In such a sociocultural context, the anonymity of online search may render it a preferable means of seeking information. (Introduction; lines 65 to 68.)

The period from 2022 to 2024 was selected based on considerations of data availability and stability. (Methods; lines 87 to 88.)

3. (L75) "the term “Utsu-byou” (the Japanese term for MDD)"

Japanese queries have common orthographic variants (e.g., うつ病 vs 鬱病) and may also be used colloquially without implying a clinical diagnosis. Please clarify whether variants were tested and/or included, and consider sensitivity analyses comparing results across key variants. In addition, because “Utsu-byou” searches may include substantial noise, please discuss how this could affect the findings and, if feasible, provide the exact Japanese search terms used (e.g., in a supplement).

Response:

As you have indicated, we have clarified in the Methods that utsu-byo refers to the Japanese term “うつ病,” In addition, we have added an explanation in the Discussion section regarding orthographic variations specific to the Japanese language and the potential bias arising from such variations as a study limitation.

Changes:

To investigate trends in online searches related to MDD, we obtained the number of search queries containing the term “うつ病” (Utsu-byo, Depression; Major Depressive Disorder) from Yahoo! JAPAN for the period from January 2022 to December 2024. (Methods; lines 85 to 87.)

Eighth, there is a possibility of bias due to orthographic variants. In Japanese, “Utsu” can be written in both hiragana and kanji, resulting in two forms, “うつ” and “鬱”, and in this study we adopted the form “うつ”. The kanji character “鬱” was added to the list of commonly used kanji in 2010 [36]. In practice, however, the form “うつ病” is more widely used in everyday contexts and is more familiar to the general public. The Japanese version of the DSM-5 and DSM-5-TR also adopts “うつ病”, indicating that it is the standard term in medical settings [13]. For these reasons, we selected “うつ病” to ensure terminological and clinical consistency. Variations in orthography may be associated with differences in demographic characteristics, such as gender or age group. However, because the data obtained in this study included only queries containing “うつ病”, potential differences related to alternative forms were not examined and should be addressed in future research. (Discussion; lines 303 to 313.)

4. (L76) "Yahoo! JAPAN is the most frequently used search engine in Japan,"

This statement should be carefully verified and appropriately framed. Metrics such as site visits to yahoo.co.jp (a large portal including news, weather, mail, etc.) are not equivalent to search engine market share. If the intent is to justify platform representativeness, please use a metric directly relevant to search usage (e.g., search engine market share) and revise the wording accordingly. Please cite an appropriate source.

Ideally, provide a supplementary table listing the symptom terms (and representative raw queries) with final ICPC-2 labels and DSM alignment.

Response:

We appreciate the reviewer’s suggestion. We changed the metric for Yahoo! usage from the number of visits to yahoo.co.jp to its search engine market share and added information on highlighting Yahoo!’s advantages. We note, however, that the data and analyses obtained through DS.INSIGHT are subject to contractual restrictions and cannot be shared with third parties, and therefore cannot be provided in this study.

Changes:

Yahoo! JAPAN has maintained the second-largest market share among search engines in Japan for many years, following Google [14]. Yahoo! provides data on the top 500 queries containing specific terms, along with information on the gender and age group of the users who performed these searches. On the other hand, Google Trends, which retrieves data from Google, provides only the regional interest trends for each query, the top 50 related queries, and the top 50 queries with rising search volume. In addition, web audience measurement data indicate that Yahoo! JAPAN continues to be widely used in Japan, with approximately 67.7% of the population using the service at least once per month in 2024, compared with 68.8% for Google during the same period [15,16]. (Methods; lines 94 to 102.)

5. (L88) "Search queries used in the analysis"

⇒Without a codebook (decision rules) and examples, readers cannot evaluate how raw queries were mapped to symptom terms and then to ICPC-2 categories—especially for slang, spelling variants, multi-term queries, and ambiguous expressions.

Moreover, because two coders independently classified terms with third-party adjudication, please report inter-rater reliability (e.g., Cohen’s kappa and/or percent agreement) and the proportion of disagreements requiring adjudication.

Response:

Thank you for this helpful comment. To improve transparency of the classification process, we have added additional methodological details in the revised manuscript, including information on inter-rater reliability (Cohen’s kappa) and examples illustrating how search queries with spelling variations or colloquial expressions were categorized.

Changes:

Representative classification rules were applied; for example, both “Depression can’t sleep” and “Depression just sleeping all day” were classified under “Sleep disturbance”, allowing for consistent categorization across slang and variant expressions. As a representative example, Cohen’s kappa coefficient was calculated for 93 symptom queries from 2024, yielding κ = 0.751, indicating substantial inter-rater agreement and sufficient reproducibility of the classification. (Methods; lines 136 to 141.)

6. (L173) "Table 3. Top 20 search terms and symptoms in online searches related to major depressive disorder by year (2022–2024)."

Please also report proportions to enable valid comparisons across years.

Response:

We have reflected this comment by adding the proportion of each query’s search volume relative to the searches for “Depression” for each year in Table 3.

Changes:

Table 3

7. (L221) "Given that the diagnostic criteria of the DSM-5-TR are based on a statistical classification of clinically observed symptoms and signs, individuals searching online for

symptoms of MDD may include patients diagnosed with MDD."

⇒Search volume reflects information-seeking behavior rather than clinical status; therefore, findings should be interpreted as public interest in symptom-related concepts linked to “Utsu-byou,” not as evidence of symptom prevalence or diagnosis. Please revise the wording accordingly to avoid potential misunderstanding.

Response:

Thank you for your important assessment. Although we have noted in the limitation that caution is needed regarding the interpretation of searchers, to avoid the potential misunderstanding that all searchers are patients with depression, we have removed the cited sentence and the following sentence, and revised the Discussion accordingly.

Changes:

This finding suggests that public awareness of depressive symptoms may be partially aligned with clinical diagnostic concepts. (Discussion; lines 249 to 251.)

[Deleted] Given that the diagnostic criteria of the DSM-5-TR are based on a statistical classification of clinically observed symptoms and signs, individuals searching online for symptoms of MDD may include patients diagnosed with MDD. For example, in cases in which patients have already been diagnosed with MDD, they may be skeptical about the doctor’s diagnosis and explanation or feel dissatisfaction and anxiety when the diagnosis is perceived as inappropriate or when treatment fails to alleviate symptoms, leading patients to seek to confirm the diagnosis or obtain further information about MDD symptoms online. Moreover, in cases in which MDD has not been formally diagnosed by a health professional, some individuals may suspect that they or an associate have MDD based on the symptoms and conduct online searches to obtain further information. In addition, in cases of severe MDD, in which psychomotor retardation and loss of motivation are common, those around affected individuals may be more likely than the affected individuals themselves to carry out searches on MDD symptoms.

Minor comments:

1. (L83) "https://datasolution.yahoo-net.jp/view/knowledgebase/641."

If this resource requires a login or is not publicly accessible, it may limit reproducibility for readers. Please provide a publicly accessible reference, or alternatively describe the relevant details in the manuscript or a supplementary appendix.

Response:

Along with your comments, we have updated the URL to a publicly accessible one.

Changes:

The user manual is available at https://datasolution.yahoo-net.jp/view/knowledgebase/2205. (Methods; lines 108 to 109.)

Others

We also indicated other minor corrections in yellow markers (please see “Revised Manuscript with Track Changes”).

We thank the reviewer for such pertinent comments. We believe that the revised manuscript adequately addresses the reviewers’ comments and has been substantially improved compared to the previous version. We hope that it is now suitable for publication in PLOS ONE.

RESPONSES TO REVIEWER 2:

We sincerely thank the reviewer for their insightful comments, which have greatly contributed to improving our manuscript. We believe that the revised version addresses all the concerns raised and represents a significant improvement over the previous submission. We hope that it is now suitable for publication in PLOS ONE.

NOTE

We highlighted changes of significant issues in yellow (please see “Revised Manuscript with Track Changes”).

Comments:

The manuscript “Online Search Interest in Major Depressive Disorder: Infodemiology Study Using the Most Visited Search Engine in Japan” sets out to characterize public interest in Major Depressive Disorder (MDD) in Japan using Yahoo! search trends from 2022 to 2024. The authors compare the most frequent terms co-searched with queries for major depressive disorder (MDD) to official classifications of MDD symptoms. Internet search data offers promising opportunities to measure public interest and public health, and we still need a clearer understanding of when and how researchers can meaningfully apply these data. I applaud the authors for advancing our understanding in this direction. However, I have two major concerns with the paper at hand regarding (1) the study goal and relevance, (2) the study’s empirical approach, and (3) the presentation of results. I elaborate on these points, along with several more minor comments, below. I hope that my comments are constructive and helpful to the authors in further strengthening their research.

Response:

Thank you for your general comments. In accordance with your suggestions, we have revised the manuscript. We have addressed each of the points you raised.

Major comments:

(1) Study goal and relevance: The authors should state the study’s research question, objectives, and scientific and societal relevance more clearly from the outset. In the introduction, the authors note that major depressive disorder (MDD) is frequently underdiagnosed or diagnosed with delay and state that the study aims “to characterize public interest in MDD” (p. 3, line 68). However, it remains unclear how the goal of characterizing public interest in MDD relates to the societal problem of delayed MDD diagnoses. It is also not specified in whic

---

## [Decision Letter · Decision Letter 1]

6 Apr 2026

PONE-D-25-64954R1Online Search Interest in Major Depressive Disorder: Infodemiology Study Using the Most Visited Search Engine in JapanPLOS One

Dear Dr. Ishizuka,

Thank you for submitting your manuscript to PLOS ONE. After careful consideration, we feel that it has merit but does not fully meet PLOS ONE’s publication criteria as it currently stands. Therefore, we invite you to submit a revised version of the manuscript that addresses the points raised during the review process.

Please address the remaining points raised by Reviewer 2

**Summary**

The reviewer 2 appreciates revisions but raises four persistent concerns about the conceptual link between public search interest and clinical diagnostic practices:

**Missing mechanistic explanation** — It's unclear how societal perceptions translate into or influence clinical diagnostics; this intermediate step needs explicit articulation.**Ambiguous implications** — The conclusion's claim about helping physicians "identify" patients needs clarification: does it mean asking more proactively about searched symptoms? If so, the contribution is about *improving* diagnostics, not just *understanding* them — a meaningful distinction.**Underdiagnosis rationale** — If the relevant symptoms are already in diagnostic catalogues, why are they being missed? The paper should explain current causes of underdiagnosis and how search interest data could address them.**Early diagnosis gap** — People searching these terms may never see a physician, so it's unclear how the findings support earlier diagnosis.

We look forward to receiving your revised manuscript.

Kind regards,

Rei Akaishi

Academic Editor

PLOS One

Journal Requirements:

Additional Editor Comments:

Thanks for submitting the new version of the manuscript and your effort to improve it. However, the reviewer 2 raise some remaining points, which seem to be the core of the issues of this paper:

Reviewer 2 Comment Summary

The reviewer appreciates revisions but raises four persistent concerns about the conceptual link between public search interest and clinical diagnostic practices:

1. Missing mechanistic explanation — It's unclear how societal perceptions translate into or influence clinical diagnostics; this intermediate step needs explicit articulation.

2. Ambiguous implications — The conclusion's claim about helping physicians "identify" patients needs clarification: does it mean asking more proactively about searched symptoms? If so, the contribution is about improving diagnostics, not just understanding them — a meaningful distinction.

3. Underdiagnosis rationale — If the relevant symptoms are already in diagnostic catalogues, why are they being missed? The paper should explain current causes of underdiagnosis and how search interest data could address them.

4. Early diagnosis gap — People searching these terms may never see a physician, so it's unclear how the findings support earlier diagnosis.

Bottom line: The discussion needs a concrete, practical explanation of what clinicians can actually do with the finding that public awareness partially aligns with clinical diagnostic concepts.

Please address these points very carefully.

Reviewers' comments:

Reviewer's Responses to Questions

**Comments to the Author**

1. If the authors have adequately addressed your comments raised in a previous round of review and you feel that this manuscript is now acceptable for publication, you may indicate that here to bypass the “Comments to the Author” section, enter your conflict of interest statement in the “Confidential to Editor” section, and submit your "Accept" recommendation.

Reviewer #1: All comments have been addressed

Reviewer #2: (No Response)

2. Is the manuscript technically sound, and do the data support the conclusions?

Reviewer #1: Yes

Reviewer #2: Yes

3. Has the statistical analysis been performed appropriately and rigorously? 

Reviewer #1: Yes

Reviewer #2: N/A

4. Have the authors made all data underlying the findings in their manuscript fully available?

Reviewer #1: No

Reviewer #2: No

5. Is the manuscript presented in an intelligible fashion and written in standard English?

Reviewer #1: Yes

Reviewer #2: Yes

6. Review Comments to the Author

Reviewer #1: (No Response)

Reviewer #2: I thank the authors for considering my earlier comments in their revision of the manuscript “Online Search Interest in Major Depressive Disorder: Infodemiology Study Using the Most Visited Search Engine in Japan”. I especially appreciate the careful revision of the presentation and discussion of the results. I see most points implemented very well which leaves me with only minor remarks that I outline in detail below. I hope my comments are constructive and helpful to the authors in further strengthening their manuscript.

(1) Relevance and goal (e.g., lines 34ff. and 50f. in the abstract, line 80f. in the introduction, line 325f. in the conclusion):

I highly appreciate the changes made in the manuscript further clarifying the study’s goal and relevance. However, I still find it difficult to understand how diagnostic practices and the measurement of public interest in MDD are conceptually linked. Specifically, it remains unclear to me how societal perceptions translate into or influence clinical diagnostic practices. I am still missing an explicit explanation of this intermediate step.

In the conclusion, the authors state that “This line of research may help primary care physicians to proactively identify patients with MDD by focusing on these commonly searched symptoms” (line 325f.). This statement makes the connection most clear compared to the abstract and introduction. However, what does identify mean in this context? Does this imply that practitioners should more proactively ask about the most searched symptoms? If so, the study’s implications are less about improving our understanding of diagnostic practices but may instead improve diagnostic practices.

At the same time, I am still wondering whether practitioners are not already asking about these symptoms anyways, given that they are listed in the clinical diagnostic catalogue. What are currently the reasons for the underdiagnosis mentioned in line 70f. and how could knowing search interest in specific symptoms help to overcome them?

Additionally, it remains unclear how the study contributes to early diagnosis if individuals searching for these terms may never consult a physician who could diagnose MDD. This gap in the argument should also be addressed in the manuscript.

Put differently, the discussion section would benefit from a clearer explanation of what practitioners can concretely learn from the finding that “public awareness of depressive symptoms may be partially aligned with clinical diagnostic concepts” (line 250f.).

(2) On the manual query classification of symptoms:

I also very much appreciate the clarifications to the manual coding process of the queries. However, I still have several questions and uncertainties regarding the coding process:

• I fully understand that the data cannot be shared due to the policies of Yahoo! JAPAN Corporation. I do, however, think that the authors need to share a codebook in the Appendix showing the coding rules for each symptom and a (synthetic) query example when a category was coded. Even if the data cannot be shared, readers need to be able to understand when a query was coded as a specific symptom.

• What are the representative classification rules the authors refer to in “Representative classification rules were applied” (line 136f.)? Providing the codebook would also help here in understanding based on which coding rules the queries were coded. The authors should also state how they developed the classification rules (i.e., iteratively and deductively).

• How did the authors choose the 93 queries that were double coded by two researchers? I assume that they have not been randomly selected as they are all from the year 2024 (which is also why I would not use the term representative in this context as in “representative example” in line 139).

Minor remarks:

Line 120: It would be great to add here whether the “Communications Usage Trend Survey” is a probabilistic or non-probabilistic survey.

Line 115: In “The internet user volume was defined as the number of internet users in Japan”, it might make the definition even more clear to specify “the absolute number of internet users in Japan”.

Line 197: The dot in “Depression test.” should come after the quotation mark.

7. PLOS authors have the option to publish the peer review history of their article (what does this mean?). If published, this will include your full peer review and any attached files.

Reviewer #1: No

Reviewer #2: No

---

## [Author Response · Author response to Decision Letter 2]

16 Apr 2026

April 16, 2026

Dear Dr. Akaishi,

Title: Online Search Interest in Major Depressive Disorder: Infodemiology Study Using the Most Visited Search Engine in Japan

Reference number: PONE-D-25-64954R1

Dear Dr. Akaishi,

Thank you for your e-mail of April 6, 2026, regarding our manuscript, “Online Search Interest in Major Depressive Disorder: Infodemiology Study Using the Most Visited Search Engine in Japan”, and for the valuable comments of the reviewer. I have attached our revised manuscript, as well as a point-by-point response to the reviewer’s comments.

We hope that the revised manuscript contains suitable responses to the comments, and we think that it has been significantly improved over the previous submission. We trust that our manuscript is now suitable for publication in PLOS ONE.

Thank you in advance for your kind consideration of our work.

Sincerely yours,

Kosuke Ishizuka, MD, PhD

Department of General Medicine, Yokohama City University School of Medicine

3-9 Fukuura, Kanazawa-ku, Yokohama-city, Kanagawa pref. Japan

Tel. +81-45-787-2800

Fax. +81-45-350-2728

E-Mail: e103007c@yokohama-cu.ac.jp

RESPONSES TO REVIEWER 2:

We sincerely appreciate the reviewer’s thoughtful comments, which have greatly contributed to improving our manuscript. We believe that the revised version has adequately addressed all the points raised and represents a substantial improvement over the previous submission. We hope that the manuscript is now suitable for publication in PLOS ONE.

NOTE

We highlighted changes of significant issues in yellow (please see “Revised Manuscript with Track Changes”).

Comments:

I thank the authors for considering my earlier comments in their revision of the manuscript “Online Search Interest in Major Depressive Disorder: Infodemiology Study Using the Most Visited Search Engine in Japan”. I especially appreciate the careful revision of the presentation and discussion of the results. I see most points implemented very well which leaves me with only minor remarks that I outline in detail below. I hope my comments are constructive and helpful to the authors in further strengthening their manuscript.

Response:

We appreciate your overall comments. The manuscript has been revised in line with your suggestions, and all issues raised have been carefully addressed.

Major remarks:

1. Relevance and goal (e.g., lines 34ff. and 50f. in the abstract, line 80f. in the introduction, line 325f. in the conclusion):

I highly appreciate the changes made in the manuscript further clarifying the study’s goal and relevance. However, I still find it difficult to understand how diagnostic practices and the measurement of public interest in MDD are conceptually linked. Specifically, it remains unclear to me how societal perceptions translate into or influence clinical diagnostic practices. I am still missing an explicit explanation of this intermediate step.

In the conclusion, the authors state that “This line of research may help primary care physicians to proactively identify patients with MDD by focusing on these commonly searched symptoms” (line 325f.). This statement makes the connection most clear compared to the abstract and introduction. However, what does identify mean in this context? Does this imply that practitioners should more proactively ask about the most searched symptoms? If so, the study’s implications are less about improving our understanding of diagnostic practices but may instead improve diagnostic practices.

At the same time, I am still wondering whether practitioners are not already asking about these symptoms anyways, given that they are listed in the clinical diagnostic catalogue. What are currently the reasons for the underdiagnosis mentioned in line 70f. and how could knowing search interest in specific symptoms help to overcome them?

Additionally, it remains unclear how the study contributes to early diagnosis if individuals searching for these terms may never consult a physician who could diagnose MDD. This gap in the argument should also be addressed in the manuscript.

Put differently, the discussion section would benefit from a clearer explanation of what practitioners can concretely learn from the finding that “public awareness of depressive symptoms may be partially aligned with clinical diagnostic concepts” (line 250f.).

Response:

We thank the reviewer for this valuable comment. After careful consideration, we have revised the study objective to make it more appropriate. We have also added an explanation in the Introduction regarding the fact that depression may remain undiagnosed even among patients who seek medical care. Furthermore, we have clarified in the Abstract, Discussion, and Conclusion how societal perceptions could contribute to identifying patients who have already accessed healthcare services but remain undiagnosed.

Changes:

- This study aimed to describe online search trends on MDD in Japan and investigate the relationship between societal perceptions of depression and the symptoms constructs in clinical diagnosis. (Abstract; lines 34 to 36.)

- Keeping in mind symptoms that attracted high attention online during clinical interviews may contribute to more proactive detection of MDD in primary care settings. (Abstract; lines 50 to 52.)

- In a multinational study of primary care facilities across 15 countries, 45% to 95% of patients with MDD presented with somatic symptoms [10]. Among patients with MDD and anxiety disorders, physicians recognized 77% of those who presented with psychosocial symptoms, but only 22% of those who presented with somatic symptoms alone. [11]. (Introduction; lines 73 to 77.)

- Additionally, we sought to investigate the relationship between societal perceptions of depression and the symptoms constructs in clinical diagnosis. (Introduction; lines 86 to 88.)

- Being mindful that not only commonly reported physical symptoms but also symptoms that attracted high levels of attention online are likely to be widely recognized by the public could help primary care physicians conduct more effective clinical assessments and improve the identification of MDD. Future research should investigate the relationship between the presence of depression or help-seeking behaviors and search activity to further strengthen these interpretations. (Discussion; 283 to 288.)

- Keeping in mind symptoms that attracted high attention online during clinical interviews may contribute to more proactive detection of MDD in primary care settings. Further research is needed to determine the extent to which such behavior reflects clinical condition. (Conclusion; 335 to 338.)

2. On the manual query classification of symptoms:

I also very much appreciate the clarifications to the manual coding process of the queries. However, I still have several questions and uncertainties regarding the coding process:

• I fully understand that the data cannot be shared due to the policies of Yahoo! JAPAN Corporation. I do, however, think that the authors need to share a codebook in the Appendix showing the coding rules for each symptom and a (synthetic) query example when a category was coded. Even if the data cannot be shared, readers need to be able to understand when a query was coded as a specific symptom.

• What are the representative classification rules the authors refer to in “Representative classification rules were applied” (line 136f.)? Providing the codebook would also help here in understanding based on which coding rules the queries were coded. The authors should also state how they developed the classification rules (i.e., iteratively and deductively).

• How did the authors choose the 93 queries that were double coded by two researchers? I assume that they have not been randomly selected as they are all from the year 2024 (which is also why I would not use the term representative in this context as in “representative example” in line 139).

Response:

Thank you for your important comment. We have added a codebook for the classification of symptom-related search queries based on ICPC-2 as S1 Table. Accordingly, we revised the Methods. In addition. We clarified that the 93 symptom queries from 2024 used to calculate Cohen’s kappa represent all symptom queries for that year.

Changes:

- For example, both “Depression can’t sleep” and “Depression just sleeping all day” were classified under “Sleep disturbance”, allowing for consistent categorization across slang and variant expressions. (Methods; lines 143 to 145.)

- As a representative example, Cohen’s kappa coefficient was calculated for all 93 symptom queries from 2024, yielding κ = 0.751, indicating substantial inter-rater agreement and sufficient reproducibility of the classification. (Methods; lines 145 to 148.)

- A detailed codebook for the classification of search queries into ICPC-2 categories is provided in S1 Table. (Methods; lines 148 to 149.)

Minor remarks:

1. Line 120: It would be great to add here whether the “Communications Usage Trend Survey” is a probabilistic or non-probabilistic survey.

Response:

Along with your comments, we have added a statement that the “Communications Usage Trend Survey” is a probabilistic survey.

Changes:

- The “Communications Usage Trend Survey” is a probabilistic survey conducted to assess trends in telecommunications usage, included 39,577 respondents in 2022, 34,196 in 2023, and 37,058 in 2024 [20-22]. (Methods; lines 126 to 129.)

2. Line 115: In “The internet user volume was defined as the number of internet users in Japan”, it might make the definition even more clear to specify “the absolute number of internet users in Japan”.

Response:

In response to the reviewer’s comment, we revised the Methods.

Changes:

- The internet user volume was defined as the absolute number of internet users in Japan. (Methods; line 122.)

3. Line 197: The dot in “Depression test.” should come after the quotation mark.

Response:

We appreciate the reviewer’s careful reading. We have corrected the typographical error.

Changes:

- The second and third most common queries in each year were “Depression” or “Depression test”. (Results; lines 204 to 205.)

We appreciate the reviewer for such helpful comments. We believe that the revised manuscript has been improved through these revisions and that the reviewer’s concerns have been addressed. We hope that the revised version is now suitable for publication in PLOS ONE.

---

## [Decision Letter · Decision Letter 2]

26 Apr 2026

Online Search Interest in Major Depressive Disorder: Infodemiology Study Using the Most Visited Search Engine in Japan

PONE-D-25-64954R2

Dear Dr. Ishizuka,

We’re pleased to inform you that your manuscript has been judged scientifically suitable for publication and will be formally accepted for publication once it meets all outstanding technical requirements.

Kind regards,

Rei Akaishi

Academic Editor

PLOS One

Additional Editor Comments (optional):

The authors have addressed the remaining concerns sufficiently. The addition of the codebook substantially improves transparency. The revised framing of the clinical implication is more cautious and acceptable, although the argument would benefit from tightening to specify that online search data may help clinicians understand patient-recognized symptom framings rather than directly improve diagnostic practice. I recommend acceptance after minor copyediting.

Reviewers' comments:

Reviewer's Responses to Questions

**Comments to the Author**

1. If the authors have adequately addressed your comments raised in a previous round of review and you feel that this manuscript is now acceptable for publication, you may indicate that here to bypass the “Comments to the Author” section, enter your conflict of interest statement in the “Confidential to Editor” section, and submit your "Accept" recommendation.

Reviewer #2: All comments have been addressed

2. Is the manuscript technically sound, and do the data support the conclusions?

Reviewer #2: Yes

3. Has the statistical analysis been performed appropriately and rigorously? 

Reviewer #2: Yes

4. Have the authors made all data underlying the findings in their manuscript fully available?

Reviewer #2: Yes

5. Is the manuscript presented in an intelligible fashion and written in standard English?

Reviewer #2: Yes

6. Review Comments to the Author

Reviewer #2: I would like to thank the authors once again for carefully addressing my previous comments in their revised manuscript. I appreciate the thoughtful consideration given to all of my points. As a minor suggestion, the inclusion of synthetic examples illustrating when a category has been coded would have further strengthened the codebook. That said, I understand that there may be constraints on what they can share. I wish the authors continued success in their research.

7. PLOS authors have the option to publish the peer review history of their article (what does this mean?). If published, this will include your full peer review and any attached files.

Reviewer #2: No

---

## [Editor Report · Acceptance letter]

PONE-D-25-64954R2

PLOS One

Dear Dr. Ishizuka,

I'm pleased to inform you that your manuscript has been deemed suitable for publication in PLOS One. Congratulations! Your manuscript is now being handed over to our production team.

Kind regards,

on behalf of

Dr. Rei Akaishi

Academic Editor

PLOS One